

# Non-gaussianity of the critical 3d Ising model

**Slava Rychkov[1,2], David Simmons-Duffin[3,4] and Bernardo Zan[5,1]**

**1** CERN, Theoretical Physics Department, 1211 Geneva 23, Switzerland
**2** Laboratoire de Physique Théorique de l'École Normale Supérieure,
PSL Research University, CNRS, Sorbonne Universités, UPMC Univ. Paris 06,
24 rue Lhomond, 75231 Paris Cedex 05, France
**3** School of Natural Sciences, Institute for Advanced Study, Princeton,
New Jersey 08540, USA
**4** California Institute of Technology, Pasadena, California 91125, USA
**5** Institut de Théorie des Phénomènes Physiques, EPFL, CH-1015 Lausanne, Switzerland

## Abstract

We discuss the 4pt function of the critical 3d Ising model, extracted from recent conformal bootstrap results. We focus on the non-gaussianity $Q$ – the ratio of the 4pt function to its gaussian part given by three Wick contractions. This ratio reveals significant non-gaussianity of the critical fluctuations. The bootstrap results are consistent with a rigorous inequality due to Lebowitz and Aizenman, which limits $Q$ to lie between $1/3$ and $1$.


[doi:10.21468/SciPostPhys.2.1.001](.)

Recent progress in the conformal bootstrap has dramatically improved our knowledge about the critical point of the 3d Ising model. The leading critical exponents are now known with $10^{-6}$ accuracy. Scaling dimensions of a dozen operators appearing in the operator product expansion (OPE) of the leading scalars are also precisely known, together with their OPE coefficients.

One interesting observable in the Ising model is the four point (4pt) correlation function of the spin field $\sigma(x)$. In the continuum limit and at the critical point, this 4pt function is constrained by conformal invariance to have the form[1]

$$\langle \sigma_1 \sigma_2 \sigma_3 \sigma_4 \rangle = \frac{g(u,v)}{|x_1 - x_2|^{2\Delta_\sigma}|x_3 - x_4|^{2\Delta_\sigma}}, \tag{1}$$

where $\Delta_\sigma = 0.5181489(10)$ [1] is the scaling dimension of $\sigma$ and $u, v$ are the conformally invariant cross-ratios: $u = (x_{12}^2 x_{34}^2)/(x_{13}^2 x_{24}^2)$, $v = u|_{1\leftrightarrow 3}$, $x_{ij} \equiv x_i - x_j$. The function $g(u,v)$ can be expanded in conformal blocks. This expansion, which will be reviewed below, is rapidly convergent, and many initial terms in it are precisely known thanks to the above-mentioned conformal bootstrap results. As a result, the four point function in the critical 3d Ising model is known in any Euclidean kinematic configuration with a percent accuracy or better.

---

[1]We use shortened notation writing 1 instead of $x_1$, $\sigma_1$ instead of $\sigma(x_1)$ etc.

In this note we would like to use this newly acquired knowledge to study the strength of non-gaussianity of the critical 3d Ising model. Namely, we will study the quantity

$$Q(1,2,3,4) = \frac{\langle \sigma_1 \sigma_2 \sigma_3 \sigma_4 \rangle}{\langle \sigma_1 \sigma_2 \rangle \langle \sigma_3 \sigma_4 \rangle + \langle \sigma_1 \sigma_3 \rangle \langle \sigma_2 \sigma_4 \rangle + \langle \sigma_1 \sigma_4 \rangle \langle \sigma_2 \sigma_3 \rangle} , \qquad (2)$$

where in the denominator we put the "gaussian" part of the 4pt function, that is, the sum of three Wick contractions. In a gaussian theory $Q = 1$, and we would like to see how strongly $Q$ deviates from 1 in the critical 3d Ising model.

Because of conformal invariance, at the critical point $Q$ depends only on the cross-ratios $u, v$:

$$Q = \frac{g(u,v)}{1 + u^{\Delta_\sigma} + (u/v)^{\Delta_\sigma}} . \qquad (3)$$

It's also convenient to apply a conformal transformation which puts all 4 points $x_i$ into a single plane and, within this plane, assigns them to $0, z, 1, \infty$ (in this order). In these coordinates we have $u = |z|^2$, $v = |1-z|^2$ and

$$Q = \frac{g(z,\bar{z})}{1 + |z|^{2\Delta_\sigma} + (|z|/|1-z|)^{2\Delta_\sigma}} . \qquad (4)$$

We can plot $Q$ as a function of $z$ in the complex plane. $Q$ is symmetric with respect to $z \to 1-z$ ($x_1 \leftrightarrow x_3$), and $z \to 1/z$ ($x_1 \leftrightarrow x_4$). A fundamental domain with respect to these two symmetries is

$$R = \{ z \in \mathbb{C} : |z| < 1, \Re z < 1/2 \} . \qquad (5)$$

In Fig. 1 we show $Q$ in the region $R$ for the critical 3d Ising model. The 4pt function $g(z,\bar{z})$ is computed by summing over the first few conformal blocks using the latest 3d Ising CFT data reported in [1, 2], see appendix A. The salient features of $Q$ visible from this plot are:

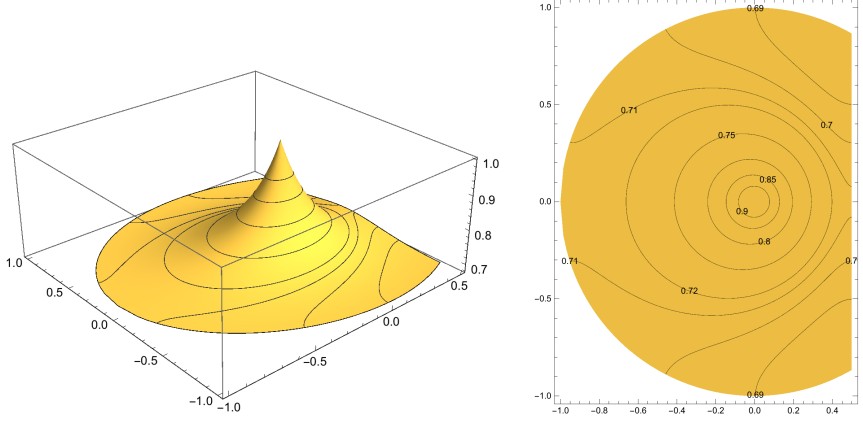

Figure 1: $Q$ in critical 3d Ising, plotted in region $R$.

1. $Q \to 1$ as $z \to 0$. This is natural since in this limit $g(u,v) \to 1$, dominated by the contribution of the unit operator in the OPE $\sigma \times \sigma$.

2. $Q$ deviates from 1 significantly. In fact, $Q < 0.75$ in a large part of $R$.

3. $Q_{\min} \approx 0.683$, attained at the two corners of the $R$ region, $z_\pm = 1/2 \pm i\sqrt{3}/2$.[2]

---

[2]For this $z$, the 4pt configuration $0, z, 1, \infty$ can be conformally mapped onto a rhombus with angles $\pi/6, 5\pi/6$. On the Riemann sphere, the same configuration can be mapped to four points equally spaced at the corners of a tetrahedron. This makes it clear why $u = v = 1$ – all the points are the same distance apart.

In comparison in Fig. 2 we show $Q$ for the critical 2d Ising model. In 2d $\Delta_\sigma = 1/8$ and the 4pt function is known exactly [3]:

$$g_{d=2}(z, \bar{z}) = \frac{|1 + \sqrt{1-z}| + |1 - \sqrt{1-z}|}{2|z|^{1/4}|1-z|^{1/4}}. \tag{6}$$

In this case $Q$ deviates even more from 1, and plateaus around 0.4 in a large portion of $R$. The minimum is

$$Q_{\min} = 1/\sqrt{6} \approx 0.408 \qquad (d=2), \tag{7}$$

attained at the same corner points $z_\pm$ as before.

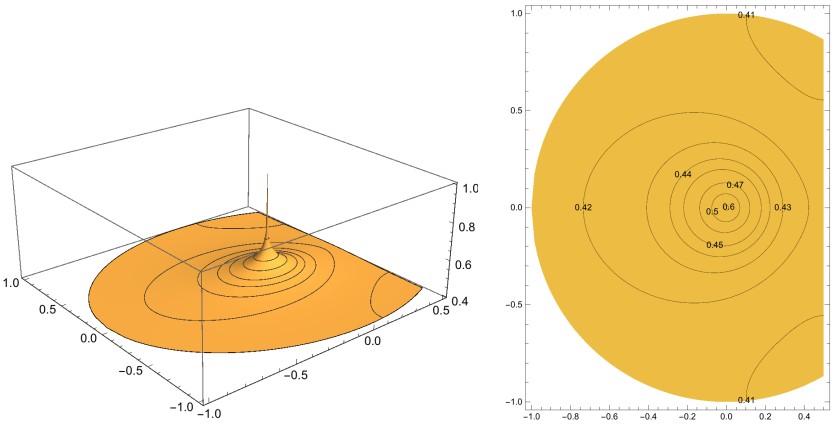

Figure 2: $Q$ in critical 2d Ising, plotted in region $R$.

On the basis of the above figures, we conclude that the critical 3d Ising model does show significant non-gaussianity. The non-gaussianity of the critical 2d model is even larger. Recall that the Wilson-Fisher fixed points interpolate between the Ising model critical points in dimensions $2 \leqslant d < 4$, becoming weakly coupled as $d \to 4$. This is compatible with the above finding.

The non-gaussianity of the 3d Ising model is a property which any attempted analytical approach to it will have to keep in mind. It's often said that the critical 3d Ising is special because the anomalous dimension of $\sigma$ is small. It is also sometimes said that it might have a weakly broken higher spin symmetry, because the higher spin currents also have a small anomalous dimension. For example, the spin 4 current anomalous dimension is 0.02274(4) [4]. However, as is clear from our study, in spite of these small anomalous dimensions, the theory does manage to deviate significantly from its gaussian approximation, so the breaking is not weak. This is certainly related to the fact that there are other operators in the theory whose anomalous dimensions are not small, of which $\epsilon$ is the prime example. It's an interesting open question if one can build an efficient approximation scheme incorporating both the sectors with small and with large anomalous dimensions. Some steps in this direction were taken in [5].

We would like to conclude this note by making contact with a curious result about the $Q$ ratio in the 3d Ising model as defined on the lattice. Namely, it can be shown that [6]

$$Q = \frac{1}{1 + 2p}, \tag{8}$$

where $p = p(1, 2, 3, 4)$ is a quantity which has probabilistic interpretation for certain curves on the lattice (closely related to high-temperature expansion graphs), so $0 \leqslant p \leqslant 1$. In particular we have:

$$1/3 \leqslant Q \leqslant 1. \tag{9}$$

The upper limit is known as the Lebowitz inequality [7], and we will call the lower limit the Aizenman inequality. For completeness, we review the derivation in appendix B. This lattice result is valid:

- in an arbitrary but finite volume with free boundary conditions,

- for any number of dimensions,

- at any temperature.

Passing to the continuum and the infinite volume limit and specializing to the critical temperature, we conclude that the same inequality (9) has to be satisfied by the critical 4pt function. It is then comforting that the plots of $Q$ given above are compatible with this two-sided inequality, in both 2d and 3d. These plots can also be seen as a prediction for the probability $p$, whose precise definition is in Eq. (B.11).

It would be interesting to find a generalization of the Lebowitz-Aizenman inequality for the $O(N)$ model. Can one find a lower bound on $Q$ which approaches 1 in the large $N$ limit? This would be natural in view of the fact that the critical point becomes weakly coupled as $N \to \infty$.

## Acknowledgement

SR is grateful to IHES where this work was initiated, and especially to Hugo Duminil-Copin for explaining the Lebowitz-Aizenman inequality. DSD thanks Clay Córdova for discussions. We thank Ulli Wolff for pointing out [15, 16]. SR and BZ are supported by the National Centre of Competence in Research SwissMAP funded by the Swiss National Science Foundation. DSD is supported by DOE grant number DE-SC0009988 and a William D. Loughlin Membership at the Institute for Advanced Study. DSD and SR are supported by the Simons Foundation grants 488655 and 488657 (Simons collaboration on the Non-perturbative bootstrap).

## A    4pt function of the 3d Ising model

In this appendix we describe the procedure for computing the conformal 4pt function $\langle \sigma\sigma\sigma\sigma \rangle$ of the critical 3d Ising model. The function $g(z,\bar{z})$ in (1) has a convergent expansion in conformal blocks:

$$g(z,\bar{z}) = \sum_{\mathcal{O} \in \sigma \times \sigma} C_{\sigma\sigma\mathcal{O}}^2 g_{\Delta_{\mathcal{O}}, \ell_{\mathcal{O}}}(z,\bar{z}), \qquad (A.1)$$

where the sum is over operators $\mathcal{O}$ in the $\sigma \times \sigma$ OPE, of dimension $\Delta_{\mathcal{O}}$ and spin $\ell_{\mathcal{O}}$. We evaluate this sum truncating to operators of dimension $< 6$, whose dimensions and OPE coefficients are reported in Table 1 of [2]. We use the same conventions and evaluation procedure for conformal blocks as in [2]. The truncation error from omitting operators of dimension $\Delta \geqslant \Delta_*$ can be estimated as [8]

$$|\delta g(z,\bar{z})| = \left| \sum_{\mathcal{O}:\Delta_{\mathcal{O}} \geqslant \Delta_*} C_{\sigma\sigma\mathcal{O}}^2 g_{\Delta_{\mathcal{O}}, \ell_{\mathcal{O}}}(z,\bar{z}) \right| \lesssim \frac{2^{4\Delta_{\sigma}}}{\Gamma(4\Delta_{\sigma}+1)} \Delta_*^{4\Delta_{\sigma}} |\rho(z)|^{\Delta_*}. \qquad (A.2)$$

Here $\rho(z) = z/(1+\sqrt{1-z})^2$ is the radial variable from [9]. In region $R$, the r.h.s. is largest at the corner points $z_{\pm}$, where $Q$ takes its minimal value. For $\Delta_* = 6$ used in this paper, estimate (A.2) gives $|\delta Q_{\min}| \sim 0.01$.

A second source of error in the 4pt function is the error in the operator dimensions and OPE coefficients as reported in [2]. We checked that this error is greatly subleading with respect to the contribution due to the operation truncation.

It is certain that the above error estimate is a huge overestimate.[3] For example, if we use the CFT data corresponding to operators of dimension $\Delta < 8$ [4], the value of $Q_{\min}$ changes only by a minuscule amount, from 0.68283 to 0.68266. It is likely that the actual variation of the 4pt function within the tiny island [1] allowed by the bootstrap constraints is of the same order as the extent of the island, which is $O(10^{-6})$. However, this has not been studied systematically and in this paper we will be content with the above estimate.

A third source of error comes from the fact that the conformal blocks are not known exactly in three dimensions. This can be made negligible expanding the blocks to sufficiently high order in $\rho$. The results reported here were obtained expanding up to order $\rho^{12}$ through a recurrence relation from [11].

## B  Lebowitz-Aizenman inequality

In this appendix we will review the proof from [6]. Consider the nearest-neighbor Ising model on a cubic lattice of a finite (but arbitrary) size. The partition function is

$$Z = \operatorname{tr} \prod_{b=\langle xy \rangle} e^{J\sigma_x \sigma_y}, \tag{B.1}$$

where $\langle xy \rangle$ denotes the bonds joining nearest-neighbor sites, and tr denotes summation over all Ising spin configurations $\sigma_x = \pm 1$. One standard way to deal with the Ising model is the high-temperature (HT) expansion, which is obtained by rewriting

$$e^{J\sigma_x \sigma_y} = (\cosh J)[1 + (\tanh J)\sigma_x \sigma_y], \tag{B.2}$$

and expanding out the product. This gives a representation of $Z$ in the form

$$Z = 2^{V_{\text{tot}}}(\cosh J)^{B_{\text{tot}}} \sum_{\eta} (\tanh J)^{B(\eta)}, \tag{B.3}$$

where the sum is over 'closed graphs' $\eta$, which are collections of bonds with the property that every vertex is touched by an even number of bonds (Fig. 3). $B(\eta)$ stands for the number of bonds in $\eta$, while $B_{\text{tot}}$ and $V_{\text{tot}}$ are the total number of bonds and vertices in the original lattice. Correlation functions like $\langle \sigma_x \sigma_y \rangle$ are then represented as a sum over graphs ending at $x, y$, etc.

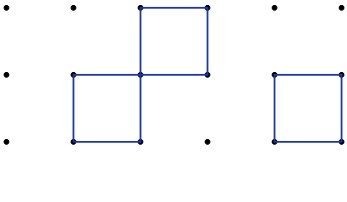

Figure 3: An example of a closed graph $\eta$.

---

[3]See [10] for a discussion of why this estimate is suboptimal, and for another estimate which becomes asymptotically better in the large $\Delta_*$ limit. For $\Delta_*$ considered here the estimate of [10] provides no significant improvement.



We will need however a different although closely related expansion. Namely, let us simply Taylor-expand the exponential, and then expand the product in (B.1). We get:

$$Z = 2^{V_{\text{tot}}} \sum_{\mathbf{n}:\partial\mathbf{n}=\varnothing} w(\mathbf{n}), \tag{B.4}$$

where $\mathbf{n}$ is a function which assigns to each bond $b$ a natural number $\mathbf{n}(b)$ – the order of the corresponding term in the Taylor expansion of the exponential. Following [6] we will call $\mathbf{n}$ a *current* (no relation to field theory currents).

The currents appearing in (B.4) have the property that the sum of $\mathbf{n}(b)$ over all bonds entering any vertex is even. The set of vertices where it's odd is denoted $\partial\mathbf{n}$. Thus the sum in (B.4) is over $\mathbf{n}$'s such that $\partial\mathbf{n} = \varnothing$ ('closed currents').

Finally, the weight of each term in the sum (B.1) is given by

$$w(\mathbf{n}) = \prod_b \frac{J^{\mathbf{n}(b)}}{\mathbf{n}(b)!}. \tag{B.5}$$

Notice that a single term in the HT expansion (B.1) sums up infinitely many terms in the current expansion. The corresponding closed currents are obtained from the closed graph $\eta$ by assigning an arbitrary positive odd number to each bond in $\eta$ and an arbitrary positive even number, or zero, to every bond not in $\eta$ (see Fig. 4). So, if the goal were to *evaluate* the partition function (or the correlation functions), it is the HT expansion that one would use. On the other hand, the current expansion may be useful for proving *relations* between correlation functions, as in the problem at hand. Indeed, since it contains many more terms than the HT expansion, there is more freedom to reshuffle those terms. These reshufflings satisfy an interesting combinatorial identity (the "switching lemma" below), which leads to relations not obvious in the HT expansions.

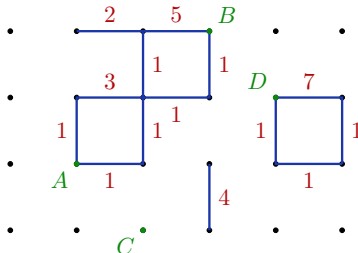

Figure 4: An example of a closed current corresponding to the closed HT graph from Fig. 3. Only the bonds for which $\mathbf{n}(b) \neq 0$ are shown. According to the definition in the text, $\mathbf{n}$ connects vertices $A, B$ but not $A, C$ or $A, D$.

We will only need a very partial case of the switching lemma. For the general case see [6, 12, 13]. A piece of notation: we will say that a current $\mathbf{n}$ *connects* a set of vertices if all these vertices belong to a single connected component of the set of bonds $b$ where $\mathbf{n}(b)$ is nonzero (see Fig. 4).

**Switching lemma (very partial case)**

$$\sum_{\partial\mathbf{n}_1=\{1,2\},\partial\mathbf{n}_2=\{3,4\}} w(\mathbf{n}_1)w(\mathbf{n}_2) = \sum_{\partial\mathbf{n}_1=\{1,2,3,4\},\partial\mathbf{n}_2=\varnothing} w(\mathbf{n}_1)w(\mathbf{n}_2)\mathbb{1}[\mathbf{n}_1+\mathbf{n}_2 \text{ connects } 3,4]. \tag{B.6}$$

Here $\mathbb{1}$ is the indicator function, which limits the summation to currents $\mathbf{n}_1$ and $\mathbf{n}_2$ satisfying the condition in brackets - namely that 3 and 4 are connect by the bondwise sum $\mathbf{n}_1 + \mathbf{n}_2$ of the two currents. Here and below we assume that all points $1, 2, 3, 4$ are distinct.

*Proof*

Let us change the summation from $\mathbf{n}_1$ and $\mathbf{n}_2$ to $\mathbf{m} = \mathbf{n}_1 + \mathbf{n}_2$ and $\mathbf{n} = \mathbf{n}_2$. Notice that

$$w(\mathbf{n}_1)w(\mathbf{n}_2) = w(\mathbf{m})\binom{\mathbf{m}}{\mathbf{n}}, \qquad \binom{\mathbf{m}}{\mathbf{n}} = \prod_b \binom{\mathbf{m}(b)}{\mathbf{n}(b)}. \tag{B.7}$$

Then

$$\text{l.h.s. of (B.6)} = \sum_{\partial\mathbf{m}=\{1,2,3,4\}} w(\mathbf{m})L(\mathbf{m}), \quad L(\mathbf{m}) = \sum_{\mathbf{n}\leqslant\mathbf{m},\partial\mathbf{n}=\{3,4\}} \binom{\mathbf{m}}{\mathbf{n}} \tag{B.8}$$

and

$$\text{r.h.s. of (B.6)} = \sum_{\partial\mathbf{m}=\{1,2,3,4\}} w(\mathbf{m})R(\mathbf{m}), \quad R(\mathbf{m}) = \mathbb{1}[\mathbf{m}\text{ connects }3,4] \sum_{\mathbf{n}\leqslant\mathbf{m},\partial\mathbf{n}=\varnothing} \binom{\mathbf{m}}{\mathbf{n}}. \tag{B.9}$$

We claim that $L(\mathbf{m}) = R(\mathbf{m})$ (from which the lemma follows). It is sufficient to consider the case when $\mathbf{m}$ connects $3, 4$, since otherwise both $L$ and $R$ are zero. We associate to $\mathbf{m}$ a graph $\mathcal{M}$ on the original lattice constructed by the following rule (see Fig. 5): replace any bond $b$ by $\mathbf{m}(b)$ parallel edges (in particular erase the bond if $\mathbf{m}(b) = 0$). For a subgraph $\mathcal{N} \subset \mathcal{M}$, we denote $\partial\mathcal{N}$ the set of vertices from which an odd number of edges originates. Then a moment's thought shows that $L(\mathbf{m})$ and $R(\mathbf{m})$ are the numbers of subgraphs $\mathcal{N} \subset \mathcal{M}$ with $\partial\mathcal{N} = \{3, 4\}$ and $\partial\mathcal{N} = \varnothing$, respectively.

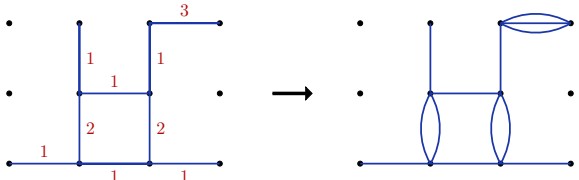

Figure 5: A current $\mathbf{m}$ with $\partial\mathbf{m} = \{1, 2, 3, 4\}$ (left) and the corresponding graph $\mathcal{M}$ (right).

Then, to show that $L(\mathbf{m}) = R(\mathbf{m})$ it's enough to exhibit a bijection between these two sets. Pick some reference subgraph $\mathcal{K} \subset \mathcal{M}$ such that $\partial\mathcal{K} = \{3, 4\}$. The crucial observation is as follows: if $\partial\mathcal{N} = \{3, 4\}$, then the symmetric difference $\mathcal{N}' = \mathcal{N}\triangle\mathcal{K}$ has $\partial\mathcal{N}' = \varnothing$ (see Fig. 6). Moreover, the map $\mathcal{N} \to \mathcal{N}'$ given by this formula is a bijection. The lemma is proven.

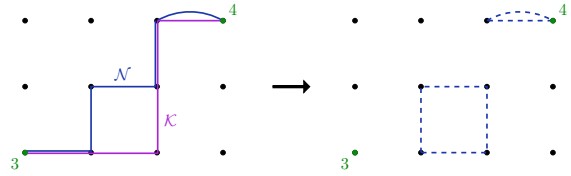

Figure 6: For the graph $\mathcal{M}$ from the previous figure, we show two subgraphs $\mathcal{N}, \mathcal{K}$ with $\partial\mathcal{N} = \partial\mathcal{K} = \{3, 4\}$ (left) and their symmetric difference $\mathcal{N}' = \mathcal{N}\triangle\mathcal{K}$ which has $\partial\mathcal{N}' = \varnothing$ (right, dashed edges).

**Derivation of the Lebowitz-Aizenman inequality**

Using the switching lemma, we will show ([6], Proposition 5.1, first line):

$$\langle\sigma_1\sigma_2\sigma_3\sigma_4\rangle - [\langle\sigma_1\sigma_2\rangle\langle\sigma_3\sigma_4\rangle + \langle\sigma_1\sigma_3\rangle\langle\sigma_2\sigma_4\rangle + \langle\sigma_1\sigma_4\rangle\langle\sigma_2\sigma_3\rangle] = -2p(1,2,3,4)\langle\sigma_1\sigma_2\sigma_3\sigma_4\rangle, \tag{B.10}$$

where

$$p(1,2,3,4) = \mathbb{P}[\mathbf{n}_1 + \mathbf{n}_2 \text{ connects } 1,2,3,4 \mid \partial \mathbf{n}_1 = \{1,2,3,4\}, \partial \mathbf{n}_2 = \varnothing] \tag{B.11}$$

$$\equiv \frac{\sum_{\partial \mathbf{n}_1 = \{1,2,3,4\}, \partial \mathbf{n}_2 = \varnothing} \mathbb{1}[\mathbf{n}_1 + \mathbf{n}_2 \text{ connects } 1,2,3,4] w(\mathbf{n}_1) w(\mathbf{n}_2)}{\sum_{\partial \mathbf{n}_1 = \{1,2,3,4\}, \partial \mathbf{n}_2 = \varnothing} w(\mathbf{n}_1) w(\mathbf{n}_2)} \tag{B.12}$$

is the probability that two currents with given boundaries satisfy the shown connectedness constraints. Eq. (B.10) is equivalent to (8), and the Lebowitz-Aizenman inequality (9) follows.[4]

Now to the derivation of (B.10). The 4pt function (normalized as it should by the partition function) is expressed in the current expansion as

$$\langle \sigma_1 \sigma_2 \sigma_3 \sigma_4 \rangle = \frac{\sum_{\partial \mathbf{n} = \{1,2,3,4\}} w(\mathbf{n})}{\sum_{\partial \mathbf{n} = \varnothing} w(\mathbf{n})} . \tag{B.13}$$

Analogously the 2pt functions $\langle \sigma_i \sigma_j \rangle$ are expressed in terms of currents with $\partial \mathbf{n} = \{i,j\}$. Multiplying the l.h.s. of (B.10) by $\left[ \sum_{\partial \mathbf{n} = \varnothing} w(\mathbf{n}) \right]^2$ we get

$$\left( \sum_{\partial \mathbf{n}_1 = \{1,2,3,4\}, \partial \mathbf{n}_2 = \varnothing} - \sum_{\partial \mathbf{n}_1 = \{1,2\}, \partial \mathbf{n}_2 = \{3,4\}} - \sum_{\partial \mathbf{n}_1 = \{1,3\}, \partial \mathbf{n}_2 = \{2,4\}} - \sum_{\partial \mathbf{n}_1 = \{1,4\}, \partial \mathbf{n}_2 = \{2,3\}} \right) w(\mathbf{n}_1) w(\mathbf{n}_2) . \tag{B.14}$$

Applying the switching lemma to each term but the first this is equal to

$$\sum_{\partial \mathbf{n}_1 = \{1,2,3,4\}, \partial \mathbf{n}_2 = \varnothing} (1 - \mathbb{1}[3 \leftrightarrow 4] - \mathbb{1}[2 \leftrightarrow 4] - \mathbb{1}[1 \leftrightarrow 4]) \, w(\mathbf{n}_1) w(\mathbf{n}_2), \tag{B.15}$$

where $\leftrightarrow$ denotes connectedness by $\mathbf{n}_1 + \mathbf{n}_2$. Now since $\partial \mathbf{n}_1 = \{1,2,3,4\}$ each point is connected to at least one other point, which implies that the points are either pairwise connected or all 4 of them are connected. In the former case the expression in brackets in (B.15) is 0, in the latter $-2$. So (B.15) can be equivalently rewritten as

$$-2 \sum_{\partial \mathbf{n}_1 = \{1,2,3,4\}, \partial \mathbf{n}_2 = \varnothing} \mathbb{1}[1,2,3,4 \text{ all connected}] w(\mathbf{n}_1) w(\mathbf{n}_2) . \tag{B.16}$$

Dividing this back by $\left[ \sum_{\partial \mathbf{n} = \varnothing} w(\mathbf{n}) \right]^2$, and then dividing and multiplying by (B.13), we obtain (B.10).

Interestingly, Eq. (B.10) can be used to perform accurate lattice Monte Carlo measurements of the $d$-dimensional Ising model 4pt function [15, 16].

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
