# Peer review of "Non-gaussianity of the critical 3d Ising model"

_SciPost Physics, doi:SciPost Phys. 2, 001 (2017)_

## Round 2 · Referee Report · Anonymous (Referee 1) · 2017-1-22

Strengths
1-The non-Gaussianity of the 3D Ising point is a subject of fundamental importance
2-The authors provide a very clear and quantitative measure of the non-Gaussianity of the 3D Ising universality class
2-The authors provide a very clear and quantitative measure of the non-Gaussianity of the 3D Ising universality class
Weaknesses
1-From a technical point of view, the results presented in this short note represent a straightforward application of previous results
Report
In this note the authors apply their previous results on the 4pt spin correlation function, obtained by conformal boostrap techniques, to measure the non-Gaussianity of the 3D Ising model. The authors presents in a very clear way the outcome of their analysis and its interpretation. I believe that the plots encoding the non-Gaussianity of 3D critical Ising model will appear on the desk of many theoretical physicists and not only.
I appreciated also the comparison of the conformal bootstrap results to rigorous inequalities obtained from the lattice model, that are nicely re-derived in the Appendix.
I appreciated also the comparison of the conformal bootstrap results to rigorous inequalities obtained from the lattice model, that are nicely re-derived in the Appendix.
Requested changes
No changes are required

---

## Round 2 · Referee Report · Anonymous (Referee 2) · 2017-1-30

Strengths
1- interesting subject 2- high precision numerical analysis 3- interesting relation with lattice results
Weaknesses
1- too concise on some technical aspects
Report
The paper under reviewal examines the four point function of spin fields in the critical 3d Ising model by employing the most recent numerical techniques and results found from through the conformal bootstrap approach.
In particular, the author studies the ratio between this four point function and the same quantity if the Wick theorem could be applied in this case. The deviation of this ratio from 1 tells about the non-gaussianity of the model.
The main result is shown in Fig. 1, which is very suggestive, given its qualitative similarity with respect to the result of the same analysis for the 2d critical Ising model. The authors also verify that their result is compatible with certain inequalities found by Lebowitz and Aizenman through a lattice analysis.
The paper is very interesting and well written. The results are presented in a clear way and I am sure that they will lead to further
insights in the analysis of the critical 3d Ising model.
I definitely support the publication of the paper in the present form.
In particular, the author studies the ratio between this four point function and the same quantity if the Wick theorem could be applied in this case. The deviation of this ratio from 1 tells about the non-gaussianity of the model.
The main result is shown in Fig. 1, which is very suggestive, given its qualitative similarity with respect to the result of the same analysis for the 2d critical Ising model. The authors also verify that their result is compatible with certain inequalities found by Lebowitz and Aizenman through a lattice analysis.
The paper is very interesting and well written. The results are presented in a clear way and I am sure that they will lead to further
insights in the analysis of the critical 3d Ising model.
I definitely support the publication of the paper in the present form.
Requested changes
no changes

---

## Editorial Decision

published